# Development of COVID-19 Vaccine Candidates Using Attenuated Recombinant Vesicular Stomatitis Virus Vectors with M Protein Mutations

**DOI:** 10.3390/v17081062

**Published:** 2025-07-30

**Authors:** Mengqi Chang, Hui Huang, Mingxi Yue, Yuetong Jiang, Siping Yan, Yiyi Chen, Wenrong Wu, Yibing Gao, Mujin Fang, Quan Yuan, Hualong Xiong, Tianying Zhang

**Affiliations:** 1State Key Laboratory of Vaccines for Infectious Diseases, Xiang An Biomedicine Laboratory, School of Public Health, Xiamen University, Xiamen 361102, China; 2National Innovation Platform for Industry-Education Integration in Vaccine Research, NMPA Key Laboratory for Research and Evaluation of Infectious Disease Diagnostic Technology, Xiamen University, Xiamen 361102, China

**Keywords:** vesicular stomatitis virus, live attenuated vaccine, SARS-CoV-2, mucosal immunity

## Abstract

Recombinant vesicular stomatitis virus (rVSV) is a promising viral vaccine vector for addressing the COVID-19 pandemic. Inducing mucosal immunity via the intranasal route is an ideal strategy for rVSV-based vaccines, but it requires extremely stringent safety standards. In this study, we constructed two rVSV variants with amino acid mutations in their M protein: rVSV-M2 with M33A/M51R mutations and rVSV-M4 with M33A/M51R/V221F/S226R mutations, and developed COVID-19 vaccines based on these attenuated vectors. By comparing viral replication capacity, intranasal immunization, intracranial injection, and blood cell counts, we demonstrated that the M protein mutation variants exhibit significant attenuation effects both in vitro and in vivo. Moreover, preliminary investigations into the mechanisms of virus attenuation revealed that these attenuated viruses can induce a stronger type I interferon response while reducing inflammation compared to the wild-type rVSV. We developed three candidate vaccines against SARS-CoV-2 using the wildtype VSV backbone with either wild-type M (rVSV-JN.1) and two M mutant variants (rVSV-M2-JN.1 and rVSV-M4-JN.1). Our results confirmed that rVSV-M2-JN.1 and rVSV-M4-JN.1 retain strong immunogenicity while enhancing safety in hamsters. In summary, the rVSV variants with M protein mutations represent promising candidate vectors for mucosal vaccines and warrant further investigation.

## 1. Introduction

The COVID-19 pandemic, caused by severe acute respiratory syndrome coronavirus 2 (SARS-CoV-2), constitutes a global health crisis [1]. Since its identification at the end of 2019, the virus has rapidly spread worldwide, severely impacting public health systems and causing widespread social and economic disruptions. The rapid development and deployment of vaccines are essential to controlling the pandemic [2]. As SARS-CoV-2 continues to mutate, scientists are developing the next generation of broad-spectrum protective vaccines [3,4], aiming to provide optimal protection for the respiratory mucosa by stimulating mucosal immunity [5,6].

Vesicular stomatitis virus (VSV), first identified in 1916 as an agricultural pathogen [7] in the United States, is known to cause outbreaks of vesicular stomatitis in pigs, cattle, and horses [8]. VSV replicates rapidly [9] and achieves high titers in most mammalian and insect cell lines [10]. Additionally, it can elicit significant humoral and cellular immune responses, making it a valuable tool in molecular virology, particularly as a vaccine vector [11]. The successful application of the VSV-vectored Ebola virus (EBOV) vaccine in 2019 [12] demonstrated the efficacy and safety [13] of VSV as an intramuscular vaccine. However, wild-type VSV is neurotropic [14] and can lead to neurological disorders in rodents and primates [15]. Moreover, the respiratory mucosa is highly sensitive and anatomically close to the central nervous system [16], making the development of mucosal vaccines [17] with high safety requirements [18] for VSV vectors highly challenging. Reducing the neurotoxicity of VSV to make it a viable candidate virus vaccine vector [19] is an urgent issue that needs to be addressed.

The neurotoxicity of wild-type VSV is primarily due to the cytotoxicity of the M protein and the neurotropism of the G protein [20]. Modifying these two proteins is a key strategy for reducing VSV’s neurotoxicity. The G protein [21], a major virulence protein of VSV, plays a critical role in viral pathogenicity. In the VSV-vectored EBOV vaccine [22], the VSV G protein was replaced by the Ebola virus glycoprotein, successfully reducing neurotoxicity while maintaining strong immunogenicity. Conversely, the M protein [23] of VSV is crucial for viral infection, promoting immune evasion and inducing cytopathic effects in host cells. Previous studies have indicated that the M33A and M51R mutations [24] in the M protein can affect its inhibition of host RNA synthesis [25], while V221F and S226R [26] mutations are associated with reduced pro-apoptotic activity and the induction of type I interferon. Yong Ke et al. [27] successfully developed a safe mucosal vaccine against SARS-CoV-2 using a highly attenuated recombinant VSV (rVSV) with three M protein amino acid mutations: S226R, V221F, and ΔM51.

In this study, we successfully constructed two rVSV variants through amino acid mutations (M33A, M51R, V221F, and S226R) in the M protein, designated as rVSV-M2 and rVSV-M4. Compared to rVSV-WT, both variants exhibited significant attenuation in vitro and in vivo. By exploring the mechanisms behind viral attenuation, we found that M protein mutations significantly enhance type I interferon responses and reduce levels of inflammatory cytokines. Furthermore, we assessed the safety of these two variants in C57/BL6J mice through intranasal administration, intracranial injection, blood cell counts, and viral pharmacokinetic distribution. More importantly, we developed three rVSV candidate vaccines against SARS-CoV-2 using the WT VSV M protein and two mutant M variants, thus confirming that viruses with M protein mutations retain strong immunogenicity while enhancing safety in hamsters. This study represents the first development of a mucosal vaccine against SARS-CoV-2 based on a highly attenuated rVSV with four M protein mutations. Given these promising results, M-mutant rVSVs, especially four amino acid mutations, potentially represent a promising candidate for a mucosal vaccine vector to combat any future COVID-19 outbreaks.

## 2. Materials and Methods

### 2.1. Cell Lines

BHK21 (American Type Culture Collection [ATCC], CCL-10), 293T (ATCC, CRL-3216), Vero (ATCC, CCL-81), A549 (CCL-185), Huh-7 (kindly gifted by Dr Chenghao Huang) were propagated in Dulbecco’s modified eagle medium (Basal Media, Xiamen, China) supplemented with 10% fetal bovine serum (Thermo Fisher Scientific, Waltham, MA, USA).

### 2.2. Generation of Mutant Plasmids

The VSV M gene fragment containing the mutation was synthesized by Sangon Biotech (Xiamen, China) and amplified by polymerase chain reaction (PCR). We constructed full-length VSV genomic plasmids containing M protein mutations (pVSV-FL-M2 and pVSV-FL-M4) by replacing the M gene in the pVSV-FL-WT plasmid with synthesized mutant fragments using XbaI and MluI (NEB, Waltham, MA, USA) restriction sites. The different fragments were purified by agarose gel electrophoresis and gel recovery (Tiangen, Beijing, China), then were connected by Gibson Assembly. GXL (Takara, Kusatsu, Japan) was used as DNA polymerase in PCR.

### 2.3. Generation of Attenuated Recombinant VSV

The 6-well plates were coated with 0.001% poly-L-lysine solution (Sigma, Waltham, MA, USA) at 37 °C with 5% CO_2_ for 30 min, then washed three times with PBS. The 293T cells were seeded into the 6-well plates at 37 °C with 5% CO_2_ for 10–12 h until 293T cells reached 80–90% confluence, infected with recombinant vaccinia virus (rVVT7, kindly gifted by Professor Juan Carlos de la Torre at Scripps Research Institute) diluted 1:3000 with DMEM. We put the 6-well plates at 37 °C with 5% CO_2_ for 1 h. Co-transfection of 3 μg pVSV-FL-M2 or pVSV-FL-M4 with helper plasmids, 0.6 μg of pBS-VSV-N, 1 μg of pBS-VSV-P, 1.6 μg of pBS-VSV-G, and 0.2 μg of pBS-VSV-L [28] were carried out with Lipofectamine^®^ LTX (Thermo Fisher Scientific) according to the manufacturer’s protocol. Recombinant vaccinia virus was discarded and washed three times with DMEM, then replaced by transfection system. The 293T cells were maintained in 10% FBS DMEM and replaced culture media approximately every 24 h. The culture media was harvested when more than 90% of cells showed cytopathic effects (CPE) and filtered through a 0.22 μm filter (Millipore, Waltham, MA, USA) into Vero cells which were transfected 5 μg pBS-VSV-G. The Vero cells were maintained in 5% FBS DMEM and replaced culture media approximately every 24 h until all cells showed more than 90% cytopathic effects (CPE). The construction process for the three COVID-19 vaccine candidates was performed following the same procedure as described above. The backbone plasmids used were pVSV-FL-JN.1, pVSV-FL-M2-JN.1, or pVSV-FL-M4-JN.1, while the helper plasmids were pBS-VSV-N, pBS-VSV-P, and pBS-VSV-L. The packaged viruses were serially passaged in BHK21-hACE2 cells.

### 2.4. Adaptive Passage and Purification of Attenuated Recombinant VSV

Rescued viruses were passaged five times in BHK-21 cells to amplify stocks, then plaque-purified in Vero cells. Vero cells were chosen for plaque purification due to their clear plaque morphology against M mutant rVSVs. A total of 70–80% of confluent Vero cells in 6-well plates were infected with serially diluted rVSV-WT, rVSV-M2, or rVSV-M4, and then covered with low melting temperature agar (Cambrex, East Rutherford, NJ, USA). Six well-isolated plaques per variant were picked, amplified, and characterized.

### 2.5. SDS-PAGE and Western Blot (WB)

Viruses were normalized to 1 × 10^6^ pfu/mL, followed by loading 30 μL per lane, separated on 12% gels (Genscript, Piscatway, NJ, USA), and either stained with Coomassie Blue or transferred to PVDF membranes. Western Blot used anti-VSV-mouse serum (1:1000) and HRP-conjugated secondary antibodies (1:5000), developed with ECL (Thermo Fisher Scientific).

### 2.6. Plaque Assay

A total of 70–80% confluent Huh-7, A549, or Vero cells in 6-well plates were infected with serially diluted rVSV-WT, rVSV-M2, or rVSV-M4, and then covered with low melting temperature agar (Cambrex). A total of 4% formaldehyde solution was used to fix the cells until plaques were visible to the naked eye. The formaldehyde solution was discarded, and the cells were covered with 1% crystal violet. Crystal violet was discarded and washed for 3 min with deionized water. Plaques were scanned and quantified with automatic intelligent imaging system EVOS M7000 (Thermo Fisher Scientific) and measured mean plaque size using ImageJ 1.8.0 software (National Institutes of Health, Bethesda, MD, USA).

### 2.7. Real-Time Fluorescence Quantitative Reverse Transcription Polymerase Chain Reaction (RT-qPCR)

The viral genome RNA was extracted using the Virus DNA/RNA Extraction Kit (GenMagBio, Beijing, China) according to the manufacturer’s protocol. RT-qPCR was performed using a forward primer (Sangon Biotech, Beijing, China), reverse primer (Sangon Biotech), probe with fluorescent labels (Sangon Biotech, Beijing, China), reverse transcriptase (TransGen, Beijing, China), and hot start version (Takara). The forward primer (TGTCTACCAAGGCCTCAAATC), reverse primer TGTATGGAGCATTAAAGGACATCCGGG), and probe (CCTGCTTTCCCGATGTTTATTC) were designed to target the VSV N gene. The entire reaction was conducted using a Gentier 96E/96R automatic medical PCR analysis system (Tianlong, Beijing, China). The viral titer was quantified using the pVSV-FL-WT plasmid at a concentration of 1 ng/μL as the standard.

### 2.8. Growth Curves

BHK21 and A549 cells were infected with rVSV-WT, rVSV-M2, or rVSV-M4, at a multiplicity of infection (MOI) of 0.0001 for 90 min at 37 °C. The culture medium was collected at 12, 24, 48, and 72 h post-infection. Virus titers (pfu/mL) were determined by plaque assay.

### 2.9. Pseudovirus-Based Neutralization Assay

In the previous work [29], we established a convenient and reliable neutralization platform based on VSV-SARS-CoV-2-S pseudovirus and hACE2-overexpressing BHK21 cells (hACE2-BHK21 cells). The JN.1 plasmid (kindly gifted by Professor Quan Yuan) was transfected into BHK21-hACE2 cells. At 48 h post-transfection, the cells were inoculated with VSVdG-EGFP-G (Addgene, Waltham, MA, USA, 31842) to allow expression of the JN.1 spike protein, followed by a 1-h incubation. The supernatant was then removed, and anti-VSV-G rat serum was added to neutralize any residual VSVdG-EGFP-G infectivity. The resulting progeny virus, VSVdG-JN.1-EGFP, was pseudotyped with the JN.1 spike protein to generate VSV pseudoviruses. Hamster serum collected at weeks 1–8 post-immunization were serially diluted and mixed with VSVdG-JN.1-EGFP pseudovirus, incubated at 37 °C for 1 h and then transferred to a 96-well plate seeded with BHK21-hACE2 cells previously. After incubation for 12 h, Opera Phenix or Operetta CLS equipment (PerkinElmer, Waltham, MA, USA) was used to obtain images and calculate fluorescence points.

### 2.10. RNA Sequencing

All the lung tissues were collected and rinsed by saline after the mice were sacrificed by CO_2_ at 12, 24, or 48 h post-infection, then placed in a sterile 1.5 mL EP tube (CellPro Bio, Beijing, China) and sent to Novegene (Beijing, China), for RNA sequencing on dry ice. The tissue block was placed directly into the mortar, and a small amount of liquid nitrogen was added. The sample was ground quickly until complete homogenization was achieved. The homogenized tissue (50–100 mg) was transferred to 1 mL of Trizol in a centrifuge tube. Chloroform accounting for 1/5 of the total volume was added, and the tube was capped tightly. The mixture was shaken vigorously for 15 s and allowed to stand at room temperature for 2–3 min. Centrifugation was performed at 12,000 rpm at 4 °C for 10–15 min. The upper aqueous phase was carefully pipetted into a new centrifuge tube, and an equal volume of isopropyl alcohol was added. The solution was mixed by inversion and left at room temperature for 10 min. Centrifugation was repeated at 12,000 rpm at 4 °C for 10 min. The supernatant was discarded, and 1 mL of 75% ethanol prepared with DEPC water was added to wash the pellet. The tube cap and walls were rinsed thoroughly, and the pellet was dislodged by flicking the tube bottom. Centrifugation was carried out at 12,000 rpm at 4 °C for 3 min, and the supernatant was removed. The pellet was air-dried at room temperature for 2–3 min. An appropriate amount of RNase-free water was added, and after complete dissolution, sequencing libraries were generated using the NEBNext Ultra RNA Library Prep Kit for Illumina (NEB, USA, Catalog: E7530L) following the manufacturer’s protocol.

### 2.11. Detection of Type I Interferon

Type I interferon was detected according to the manufacturer’s protocol after the mice serum or cell culture supernatant was optimally diluted. Huh-7 and A549 cells were infected with rVSV-WT, rVSV-M2, or rVSV-M4 at MOI of 0.0001 and supernatants were harvested at 12, 24, 48, and 72 h post-infection, with supernatants being replaced by an equal volume with 5% FBS DMEM. Cell supernatants were analyzed using a customizable Luminex kit (R&D Systems, Minneapolis, MN, USA), which allowed the multiplex detection of selected cytokines which were detected with Luminex200 (Thermo Fisher Scientific). For IFN-β detection in mouse serum, an ELISA assay (R&D Systems, Catalog: MIFNB0) was performed, followed by absorbance measurement at 450 nm with PHOMO (AutoBio, Beijing, China).

### 2.12. Animals

C57BL/6 mice were purchased from SLAC ANIMAL (Shanghai, China). LVG Golden Syrian Hamsters were purchased from Vital River (Beijing, China). All animals were housed at the National Institute of Diagnostics and Vaccine in Infectious Diseases of Xiamen University. All animal experiments were approved by the Institutional Animal Care and Use Committee at Xiamen University (Approval No.: XMULAC20240079). The viruses were diluted to the desired titer using PBS. C57BL/6 mice were immunized intranasally with 50 μL per animal. C57BL/6 mice were immunized intracranially with 20 μL per animal. Hamsters were immunized intranasally with 100 μL per animal.

### 2.13. Animal Vaccination, Infection and Tissue Harvest

Sex- and age-matched animals were randomly assigned to experimental groups. Six- to eight-week-old animals were anesthetized with isoflurane prior to vaccination or blood sampling. At experimental endpoints or humane endpoints, animals were sacrificed with excess carbon dioxide. The collected hamster lung tissues were homogenized in cold PBS using a tissue homogenizer (Servicebio, Beijing, China). Homogenate was then centrifuged at 4000× *g*. at 4 °C for 10 min. The supernatant was separated and stored at −80 °C for future assays.

### 2.14. Hematoxylin and Eosin (H&E) Staining

The animal tissues were washed twice in PBS and then fixed in 4% paraformaldehyde solution for 24 h or longer. Finally, paraffin embedding, sectioning, and staining were performed.

### 2.15. Statistics

Statistical analysis was performed using GraphPad Prism 9.4.1 (GraphPad Software, Inc., San Diego, CA, USA). A two-way ANOVA was used to analyze the statistical analysis of different groups, and *p* value <0.05 was considered as significant difference.

ID_50_ (defining the 50% infectious dose, plasma/serum dilutions causing a 50% reduction in the number of GFP-positive cells) was calculated by nonlinear regression. The average plaque size was determined by measuring the area of each plaque in each group using ImageJ 1.8.0 software (National Institutes of Health, Bethesda, MD, USA).

## 3. Results

### 3.1. Construction of Recombinant VSV with Amino Acid Mutations in M Protein

It has been reported that M33A and M51R substitutions are both associated with VSV cytotoxicity. The M51R substitution is known to affect M protein-mediated inhibition of host RNA synthesis. The plasmid pVSV-FL-M2, containing M33A and M51R mutations, was constructed through reverse genetics technology. Moreover, V221F and S226R are associated with reduced viral host shutdown, decreased pro-apoptotic activity and induction of type I interferon. The plasmid pVSV-FL-M4, containing M33A, M51R, V221F, and S226R mutations was constructed by the same means (Figure 1A). Subsequently, we constructed two attenuated rVSVs, named rVSV-M2 and rVSV-M4 through virus rescue technology (Figure 1B). These two viruses were passaged adaptively in BHK21 cells and conducted plaque purification in Vero cells to obtain monoclonal viruses. Viral stocks were amplified by infection at 0.0001 MOI in BHK21 cells. To further confirm that mutations had remained in the viral M gene, the VSV RNA was extracted. Reverse transcription PCR, agarose gel electrophoresis, purification, and next generation sequencing technology was used to verify the correctness of the viral M mutations (Figure 1C).

BHK21 cells were infected with rVSV-WT, rVSV-M2, or rVSV-M4 at a MOI of 0.0001 separately, and culture supernatants were harvested when the cells were completely cytopathic. The supernatants were subjected to SDS-PAGE and Western Blot (Figure 1D) experiments to identify the integrity of the viral proteins. The results showed that there was no significant difference in the molecular weight and expression level of M protein between M-mutant VSV and wild type VSV.

### 3.2. Comparison of the Replication Capacity of rVSV-WT, rVSV-M2, and rVSV-M4

It has been reported that the type I interferon response of M protein mutant rVSV is enhanced [30]. Therefore, BHK21, Huh-7, and Vero cells with disturbed type I interferon response, A549 cells with normal type I interferon response were chosen to characterize the properties of rVSV-WT, rVSV-M2, and rVSV-M4 at a MOI of 0.0001. BHK21 and A549 cells were infected with rVSV-WT, rVSV-M2, or rVSV-M4 at a MOI of 0.0001 and supernatants were harvested at various time points. Viral titers were quantitated by the plaque assay (Figure 2A). The results showed that the three viruses formed the same cytopathic effects (Appendix A) and the viral titers were not significantly different in BHK21 cells. However, the titer of M protein mutant rVSV in A549 cells was significantly lower than that of the rVSV-WT, suggesting that type I IFN produced by A549 cells infected with rVSV-M2 or rVSV-M4 could inhibit the viral production. Additionally, viral titers have no titer difference between the three viruses supporting the sensitivity of the M mutants to interferon.

In addition, plaque assay (Figure 2B,C) was used to compare the replication ability of rVSV-WT, rVSV-M2, and rVSV-M4. The plaque results revealed the average plaque areas of M mutant rVSVs were not significantly different from rVSV-WT in Vero (rVSV-WT: 1.831 ± 0.3808 mm^2^, rVSV-M2: 1.753 ± 0.3764 mm^2^, rVSV-M4: 1.777 ± 0.2607 mm^2^) and Huh-7 (rVSV-WT: 1.529 ± 0.3636 mm^2^, rVSV-M2: 1.697 ± 0.3076 mm^2^, rVSV-M4: 1.621 ± 0.2077 mm^2^) cells. In A549 cells, the average plaque areas produced by rVSV-WT (2.739 ± 0.3702 mm^2^) were much larger than those by rVSV-M2 (1.562 ± 0.2939 mm^2^); however, areas produced by rVSV-M2 were also larger than those by rVSV-M4 (1.017 ± 0.1278 mm^2^). These findings suggest that both rVSV-M2 and rVSV-M4 have weaker replication and appear to be more attenuated than rVSV-WT in in vitro.

### 3.3. Type I IFN Expression Enhancement by M Protein Mutant

Huh-7 and A549 cells were chosen to evaluate the interferon responses between the three rVSV viruses. The two cells were infected with rVSV-WT, rVSV-M2, or rVSV-M4 at MOI of 0.0001 and supernatants were harvested at 12, 24, 48, and 72 h post infection. IFN-α and IFN-β in the supernatants were detected using the Luminex assay, according to the manufacturer’s protocol (Figure 3A and Appendix A). The results indicated that IFN-α and IFN-β expression were below the detection limit (0.476 pg/mL) in Huh-7 cells. In A549 cells, the interferon response was found to be the highest in the rVSV-M4 group compared to the rVSV-M2 group, with the weakest response in the rVSV-WT group. The above results indicated that in cell lines with normal type I interferon response, rVSV-M2 and rVSV-M4 induced a stronger type I interferon response than rVSV-WT.

Subsequently, C57BL/6 mice were immunized intranasally with rVSV-WT, rVSV-M2, rVSV-M4, or PBS at a single dose of 1 × 10^7^ PFU/50 μL or 1 × 10^5^ PFU/50 μL. Six mice were included per group. Serum was collected at 12, 24, and 48 h post-infection. IFN-β in serum was detected through the enzyme-linked immunosorbent assay (ELISA), according to the manufacturer’s protocol (Figure 3B). At a single dose of 1 × 10^7^ PFU immunization, IFN-β was detectable in the rVSV-M2 and rVSV-M4 groups as early as 12 h post immunization, while in the rVSV-WT group it was detectable only at 24 h post immunization. However, at a single dose of 1 × 10^5^ PFU immunization, the rVSV-WT group exhibited a significantly stronger INF-β response than that of the attenuated groups. However, IFN-beta appears to be very low for the M4 virus. It was hypothesized that this was due to the reduced replication capability of the M protein mutant viruses in mice. To validate this hypothesis, another intranasal immunization of C57BL/6 mice with a single dose of 1 × 10^5^ PFU was conducted. The mice were then sacrificed, and whole lung tissue was collected for transcriptome sequencing. Sequencing results showed that at 12 h post-immunization (Figure 3C), a stronger type I interferon response was induced in the M protein mutant virus groups than in the wild type group. These findings suggest that rVSV-M2 and rVSV-M4 can induce a faster and stable innate immune response than rVSV-WT, aiding in the host defense against viral infection.

### 3.4. Safety Evaluation In Vivo

VSV has demonstrated good safety in intramuscular injection. However, higher safety standards are required for VSV when mucosal immunization vaccines were being developed. Previous studies have indicated that neurological diseases in animals could be caused by high doses of wild-type VSV. To assess the safety of M protein mutant viruses, C57BL/6 mice were immunized intranasally with rVSV-WT, rVSV-M2, or rVSV-M4 at a single dose (1 × 10^7^ PFU/50 μL, 1 × 10^5^ PFU/50 μL, or 1 × 10^3^ PFU/50 μL) diluted in PBS. The body weight was monitored to evaluate the safety of M protein mutant viruses (Figure 4A). The results showed that at ×10^5^ PFU and 1 × 10^3^ PFU dose, the weight loss of mice in the rVSV-WT group was faster and more pronounced compared to the attenuated recombinant groups, especially in the rVSV-M4 group. At 21 days post-immunization, the mice were challenged with a single dose of 1 × 10^7^ PFU rVSV-WT to determine whether protection against the rVSV-WT was provided by the attenuated recombinant viruses (Figure 4B). The results indicated that no significant weight loss was observed in any of the three groups of mice at immunization one dose of 1 × 10^7^ PFU or 1 × 10^5^ PFU. At an immunization dose of 1 × 10^3^ PFU, M protein mutant viruses were found to provide a level of protection not inferior to that of rVSV-WT. These findings suggest that the virus with the M protein mutation could still stimulate strong protection against rVSV-WT.

The intracerebral injection experiment is considered the most sensitive test to evaluate viral neurotoxicity. To assess the neurovirulence of M protein mutant viruses, C57BL/6 mice were inoculated intracranially with rVSV-WT, rVSV-M2, or rVSV-M4 at a single dose of 1 × 10^7^ PFU (Figure 4C,D) or 1 × 10^3^ PFU (Figure 4E,F). Similar to intranasal immunization, mice injected with rVSV-WT began losing weight on day 2, with more pronounced weight loss at day 7 at one dose of 1 × 10^3^ PFU. At a dose of 1 × 10^7^ PFU/20 μL, the rVSV-WT group exhibited three deaths with a survival rate of 50%, while at 1 × 10^3^ PFU/20 μL, this group showed two deaths with a 66.7% survival rate. In contrast, the attenuated variant groups demonstrated no mortality at either dose level, achieving 100% survival. At both doses, all mice challenged with M protein mutant viruses survived and the weight loss was significantly lower than that in the WT group. However, mice in rVSV-WT groups were rapidly lethal within 10 days. These results indicate that M protein mutant VSV can significantly reduce the neurotoxicity and enhance the safety.

Studies have shown that the levels of inflammatory cytokine increase after viral infection and decrease after the virus is cleared. Therefore, the levels of TNF-α, IL-6, IFN-γ, and CCL4 in serum were measured at 4, 24, and 48 h post infection (Figure 5A). The results showed that in the rVSV-WT group, the levels of TNF-α, IFN-γ, and CCL4 were significantly higher at 24 h post-infection compared to the rVSV-M2 and rVSV-M4 groups. However, the IL-6 level in the attenuated mutant group was found to be higher than that in the rVSV-WT group at 4 h post-infection, but was lower at 24 h post-infection. It was believed that the virus with the M protein mutation induced an earlier and more stable immune response.

The complete blood cell count of mice was treated as an important indicator for assessing viral safety. The number and proportion of cells in the blood were evaluated to understand the immune system’s response to the virus. Complete blood cell counts were conducted on C57BL/6 mice that were immunized intracranially with rVSV-WT, rVSV-M2, or rVSV-M4 at a single dose of 1 × 10^7^ PFU at 24 h, 48 h, 72 h, and 120 h post-infection (Figure 5B). Severe reductions in total white blood cells, neutrophils, lymphocytes, and monocytes in the blood from 24 to 72 h post-infection were observed in mice infected with rVSV-WT, particularly at 48 h post-infection when the numbers of various immune cells dropped to their lowest point. Compared to the rVSV-WT group, the blood cell counts in the rVSV-M2 and rVSV-M4 groups were observed to be similar but less severe. Specifically, at 48 h post-infection, the number of various blood cells in group rVSV-M4 was found not to be significantly different from the PBS group but prior (24 hpi) the levels were lower. The PBS control group had also experienced elevated WBC, monocytes, and lymphocytes at 120 hpi as compared to 24 hpi probably due to the IC inoculation. RBC, platelets, and neutrophils had no changes occurring. Additionally, viral infection was usually accompanied by thrombocytopenia. In the rVSV-WT group, platelet counts in the blood were drastically reduced at 24 and 48 h post-infection, whereas this was not observed in rVSV-M2 and rVSV-M4 groups. At 120 h-post infection, the platelet counts for the WT VSV group were not statistically different from the control thereby the thrombocytopenia is transient. These results indicated that the viruses with the M protein point mutation were considered less reactogenic than the rVSV-WT. These results suggest that the M protein mutant rVSVs demonstrated improved reactogenicity compared with rVSV-WT in vivo.

### 3.5. Construction of Mucosal COVID-19 Vaccine Candidates Based on rVSV

Efficacy and safety are two fundamental characteristics for an ideal vaccine vector. The aforementioned experiments demonstrated that rVSV-M2 and rVSV-M4 are safer than rVSV-WT. We constructed three COVID-19 vaccine candidates based on wild type and attenuated rVSV, named rVSV-JN.1, rVSV-M2-JN.1, and rVSV-M4-JN.1 (Figure 6A). Hamsters were immunized intranasally with a single dose of 1 × 10^5^ PFU of rVSV-JN.1, rVSV-M2-JN.1, or rVSV-M4-JN.1 and sacrificed on the second and fifth days post-immunization to obtain lung tissues for virus titer detection (Figure 6B), complete blood cell count (Figure 6C), and hematoxylin and eosin (H&E) staining. Lung histological analysis showed that the hamsters in the attenuated modification groups had reduced substantive lesions and diffuse infiltration compared to the rVSV-JN.1 group. The RNA titers of the three viruses were quantified by RT-qPCR. The results indicated that rVSV-JN.1 exhibited a higher viral load on the second day compared to the other two groups. Although the *p*-value was between 0.05 and 0.1, which is marginally significant, this observation suggests that the M protein mutant rVSVs have a slower replication rate in vivo, leading to enhanced reactogenicity compared to rVSV-JN.1. Furthermore, hematological analyses, including complete blood count assessments of hamsters at specific time points, revealed no significant differences in the counts of leukocytes, neutrophils, lymphocytes, and monocytes between the three experimental groups and the control group. Based on the aforementioned experimental results, it was concluded in this study that after replacing the VSV G protein with the SARS-CoV-2 JN.1 spike protein variant, the viral toxicity induced by the VSV G protein was significantly attenuated. Under these conditions, the viral toxicity was primarily mediated by the VSV M protein, thus no significant reactogenicity was observed in the bloodstream. Therefore, alternative detection methods were required to compare the safety profiles of rVSV-JN.1, rVSV-M2-JN.1, and rVSV-M4-JN.1 in hamsters. The results of HE staining (Appendix A) of the lung tissue showed that there was no severe pathological damage after infection with the three viruses. However, in localized areas (Figure 6D), the lung tissue of hamsters in the rVSV-JN.1 group showed significant infiltration of granulocytes in the alveolar walls, with small regions of alveolar consolidation. In contrast, in the lung tissue of the other two groups of hamsters, there was only minimal granulocyte infiltration in the alveolar walls, and the alveolar walls were slightly thickened. In addition, we conducted histological scoring (Figure 6E) on the lung tissues of hamsters in each group and presented representative HE-stained images of the lungs. The scoring results revealed that hamsters immunized rVSV-JN.1 had significantly higher scores compared to the other two groups. These results indicate that JN.1 viruses constructed using the attenuated VSV vectors had higher reactogenicity.

We next examined the antibody response in hamsters vaccinated with these three vaccines. Hamsters were vaccinated via intranasal inoculation with two doses (1 × 10^6^ PFU per dose) of rVSV-JN.1, rVSV-M2-JN.1, or rVSV-M4-JN.1 at an interval of 21 days and bled at specific time points for serum collection. JN.1 neutralizing antibodies were detected using the SARS-CoV-2 pseudovirus neutralization platform previously constructed (Figure 6F). The results demonstrated that within three weeks after the first immunization, hamsters in the rVSV-JN.1 group rapidly developed robust antibody responses, with neutralizing antibody ID_50_ reaching 10^4^ or higher. In contrast, the rVSV-M2-JN.1 group only elicited weak antibody responses, while the rVSV-M4-JN.1 group achieved neutralizing antibody ID_50_ titers of 10^3^ or above. During weeks 4–8 post-immunization, the rVSV-JN.1 group maintained strong antibody responses, with ID_50_ values ranging between 10^4^ and 10^5^. Notably, the rVSV-M2-JN.1 group developed potent antibody responses after the second immunization, reaching ID_50_ titers around 10^4^. The rVSV-M4-JN.1 group exhibited comparable antibody responses to rVSV-JN.1, with ID_50_ values within the 10^4^ to 10^5^ range. These findings collectively indicate that all three groups of hamsters were capable of generating neutralizing antibodies against JN.1 spike protein following two-dose immunization.

## 4. Discussion

Since early 2020, the COVID-19 pandemic has lasted for five years. Although SARS-CoV-2 no longer constitutes a public health emergency of international concern as defined by the WHO, over 1700 people worldwide still die from COVID-19 each week, and vaccine coverage among high-risk populations continues to decline [31]. Therefore, the development of safe and effective respiratory mucosal [32] vaccines against COVID-19 is urgent [33]. From the initial isolation of VSV in 1925 [34] to the clinical approval of the VSV-based Ebola vaccine [35] in 2019, VSV has evolved from an agricultural pathogen to a viral vector and has been proven to be an effective vaccine vector [36]. Research has indicated that a VSV-based COVID-19 vaccine [37] is more suitable for nasal administration rather than the intramuscular route. However, the development of respiratory mucosal vaccines imposes stringent safety [38] requirements on VSV vectors. In this study, we constructed two attenuated VSV vectors, rVSV-M2 and rVSV-M4, by introducing mutations in M protein of rVSV and compared the safety with rVSV-WT both in vitro and in vivo. Moreover, we took advantage of VSV M mutant vectors to construct COVID-19 vaccines, aiming to address the ongoing threat of the pandemic.

Markus Hoffmann et al. [39] previously reported the generation of M-mutant VSV and conducted preliminary explorations of its virulence, but did not utilize it as a vaccine vector. Using reverse genetics and previously developed virus packaging technology, we constructed two-point and four-point mutated viruses. In vitro, we compared the growth curves of these three viruses in BHK21 and A549 cells. In BHK21 cells, there was no significant difference in viral titers among the three viruses. However, in A549 cells, the titer of rVSV-WT was significantly higher than the other two. Similarly, when comparing viral replication capacity, rVSV-WT replicated faster in cells with a normal interferon response. These results suggest that the point mutations in the M protein could enhance the safety of the virus in vitro.

Innate immunity is the body’s first line of defense and provides rapid and broad-spectrum protection against various pathogens. Type I interferons are a critical component of the innate immune system and play a key role in antiviral immunity [40]. We compared the levels of type I interferon induced by the three viruses in the Huh-7 and A549 cell lines. Consistent with expected results, there were no significant differences in interferon levels induced by the three viruses in the Huh-7 cells. However, in the A549 cells, which have a normal type I interferon pathway, the M protein mutant rVSVs induced a stronger type I interferon response. Additionally, we compared the levels of IFN-β induced by the three viruses in C57BL/6 mice. Upon immunized with one dose of 1 × 10^7^ PFU, the M protein mutant rVSVs induced a quicker interferon response to combat viral infection, although the overall IFN-β response was significantly higher for the rVSV-WT virus at 24 h-post-infection than the M mutant rVSVs. At one dose of 1 × 10^5^ PFU, during our observation period, the IFN-β levels induced by the M protein mutant rVSVs were lower than those induced by rVSV-WT. This may suggest that amino acid mutations in M protein reduce the pathogenicity of the virus, thereby weakening the interferon response. To further validate our hypothesis, we conducted transcriptomic sequencing on the whole lung tissue of the mice. The results showed that the M protein mutant rVSVs induced a stronger interferon response at 12 h post-infection. These experimental results indicate that amino acid mutations in M protein enhance type I interferon responses both in vitro and in vivo, which helps inhibit viral replication. Moreover, many tumor cells are deficient in type I interferon signal. Therefore, the M protein mutant rVSV can replicate rapidly in tumor cells but less in normal cells, making it a promising oncolytic virus vector [34,41].

Studies have shown that vaccines administered via nasal administration can induce a relatively broad-spectrum immune response locally compared to those given through muscle injection [42]. In response to sudden public health events, such as the COVID-19 pandemic, developing mucosal immunity vaccines is crucial. VSV is a promising viral vaccine vector; however, its safety in the nervous system still needs improvement. Therefore, we compared the safety differences between two attenuated replicative viral vectors constructed in this study and the wild-type virus via intracranial injection and intranasal immunization in C57BL/6 mice. Surprisingly, a single dose of 1 × 10^7^ PFU rVSV-M2 or rVSV-M4 did not cause mortality, whereas rVSV-WT caused 20% mortality in mice at 1 × 10^3^ PFU. The results of intranasal immunization were consistent with intracranial immunization, fully demonstrating that rVSV-M2 and rVSV-M4 had significantly improved safety in terms of neurotoxicity. The results of both intranasal and intracranial immunization were consistent, fully demonstrating that rVSV-M2 or rVSV-M4 had significantly improved safety in terms of neurotoxicity. Furthermore, to evaluate the protective immune response in vivo, we first immunized C57BL/6 mice with different doses of the three viruses and then challenged them with 1 × 10^7^ PFU of rVSV-WT. The results showed that the M-mutated viruses could still provide protective effects against rVSV-WT.

Complete blood cell count is an important method for evaluating virus safety and can reflect the health status of the animals. Specifically, leukocyte differential count can provide a detailed assessment of the immune system. We performed complete blood cell counts at different time points after intranasal immunization of C57BL/6 mice with three types of viruses. We found that lymphocytes in the rVSV-WT group decreased significantly and took longer to return to normal levels. In contrast, rVSV-M2 and rVSV-M4 groups allowed lymphocytes to recover to normal levels more quickly, reducing the risk of infection and other diseases. Additionally, we evaluated differences in levels of inflammatory cytokines produced by C57BL/6 mice in response to the three viruses. Inflammatory cytokines play a key role in initiating and regulating the inflammatory response and promotes the immune system’s ability to recognize and respond to infections, injuries, or other threats through various mechanisms. We monitored cytokine levels during the first two days post-infection in C57BL/6 mice. The results showed that the rVSV-M2 and rVSV-M4 groups induced a quicker and more stable antiviral response compared to the rVSV-WT group, consistent with our previous transcriptome analysis results.

In our study, we constructed three versions of recombinant viruses by replacing the VSV G gene with the COVID-19 variants’ JN.1 spike gene and introducing two mutations in the M protein of VSV, named rVSV-M2-JN.1 and rVSV-M4-JN.1. In the hamster model, lung histological analysis and quantitative assessment of viral titers also demonstrated higher reactogenicity. Additionally, neutralizing antibody titers in infected hamster serum were measured over 8 weeks, confirming that rVSV-M2-JN.1, or rVSV-M4-JN.1 still induced high levels of neutralizing antibodies against the VSV-JN.1-Spike pseudovirus. In summary, we verified that the attenuated M protein mutant rVSVs not only maintain their efficacy but also feature superior reactogenicity. Notably, the innovatively engineered rVSV-M4 variant shows particular promise as a safer viral vaccine platform. However, our viral distribution analysis was primarily focused on lung tissue as the target site for respiratory vaccination. Future studies will include comprehensive biodistribution and viremia analyses. We have revised and corrected in the updated manuscript to ensure its appropriateness.

The VSV M protein can induce a “host shut-off” effect by inhibiting host cell transcription and preventing the transport of cellular mRNA from the nucleus to the cytoplasm. This effect effectively reduces the synthesis of critical immune molecules, such as type I interferons, by the host cell, thereby blocking the host’s antiviral response and facilitating viral replication. When specific amino acid mutations in the M protein, the host shut-off activity is weakened. In particular, the mutations V221F and S226R significantly reduce host shut-off activity. These mutations possibly alter the three-dimensional structure of the M protein, decreasing its interaction with the host cell’s transcription and RNA transport systems, thereby diminishing its pathogenic functions. Furthermore, previous studies have shown that the M51R mutation can enhance type I IFN production [24] to reduce the toxicity of VSV. Therefore, our study suggests that specific amino acid mutations in the M protein can decrease host shut-off activity and induce type I interferon production, thereby attenuating the cytotoxicity of VSV. The mutations fundamentally alter virus-host interactions by significantly potentiating the host interferon response, as evidenced by earlier IFN-β detection, sustained upregulation of interferon genes. Perhaps most crucially for vaccine development, the attenuated strains demonstrate a remarkably well-controlled inflammatory profile characterized by more balanced inflammatory cytokine responses that avoid excessive tissue damage, coupled with faster yet smoother overall antiviral reactions that facilitate an optimal transition to adaptive immunity.

Although we have mutated the M protein and verified its higher safety compared to rVSV-WT, theoretically, a replication-defective vector has superior reactogenicity and more promise as a vaccine vector [43]. However, packaging a replication-defective viral vector and achieving high titers akin to replication-competent vectors remains very challenging. In the future, we plan to conduct more in-depth research on replication-defective rVSV, including M or P gene deleted, as vaccine and oncolytic vectors. Moreover, regarding the attenuation mechanism of the M protein point mutation, we have primarily explored the interferon response and some common inflammatory cytokine responses. We will conduct further research on aspects such as trained immunity in the future.

In summary, this study constructed two types of attenuated, replication-competent rVSVs with amino acid mutations in the M protein. Both cell and animal models confirmed their higher safety compared to rVSV-WT. Moreover, we conducted a preliminary exploration of the attenuation mechanisms. Additionally, this study successfully constructed COVID-19 vaccine candidates using M-mutant rVSV vectors and validated their safety and neutralizing antibody levels in hamster models. Further validation is required to determine if these two vaccine vectors can be developed into effective respiratory mucosal vaccine vectors.

## Figures and Tables

**Figure 1 viruses-17-01062-f001:**
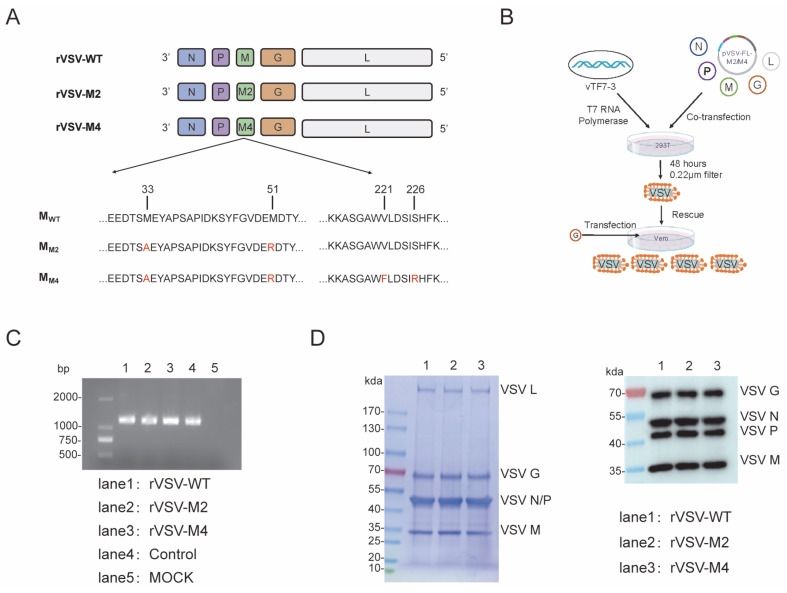
Construction of recombinant VSV with amino acid mutations in M protein. (**A**,**B**) Schematic representation of M protein mutant rVSVs construction. Plasmids with mutations in the VSV M gene were constructed using reverse genetics. Genomic plasmids (pVSV-FL-WT, pVSV-FL-M33A/M51R, or pVSV-FL-Mmut) along with helper plasmids (pBS-VSV-N, pBS-VSV-P, pBS-VSV-G, and pBS-VSV-L), were co-transfected into 293T cells pre-infected with vTF7-3. Subsequently, the pBS-VSV-G plasmid was transfected into Vero cells to rescue the virus, resulting in the construction of three viruses: rVSV-WT, rVSV-M2, and rVSV-M4. (**C**) Agarose gel electrophoresis (AGE) and next generation sequencing technology was used to verify the correctness of the viral M gene. (**D**) SDS-PAGE (left) and Western Blot (right) analyses were identified the integrity of the viral proteins.

**Figure 2 viruses-17-01062-f002:**
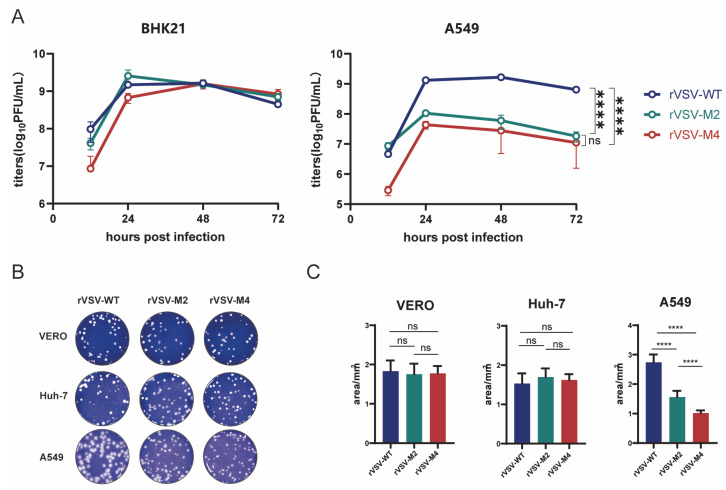
Comparison of the replication capacity of rVSV-WT, rVSV-M2, and rVSV-M4. (**A**) BHK21 or A549 cells were infected with rVSV-WT, rVSV-M2, or rVSV-M4 separately (MOI of 0.0001). Virus supernatants were harvested at 12, 24, 48, and 72 h post infection. Viral titers (PFU/mL) were determined by plaque assays. GraphPad Prism was used to analyze the significant differences using the one-way ANOVA method. ****, *p* < 0.0001; ns, *p* > 0.05. (**B**) Plaque assays were conducted to compare the replication capabilities of the three viruses in A549, Vero, and Huh7 cells. (**C**) Ten independent plaques formed by each virus were randomly selected, and plaque areas were quantified using ImageJ 1.8.0 software (National Institutes of Health, Bethesda, MD, USA). GraphPad Prism was used to analyze the significant differences in plaque size using the one-way ANOVA method. Data are presented as mean ± standard deviation (SD). ****, *p* < 0.0001; ns, *p* > 0.05.

**Figure 3 viruses-17-01062-f003:**
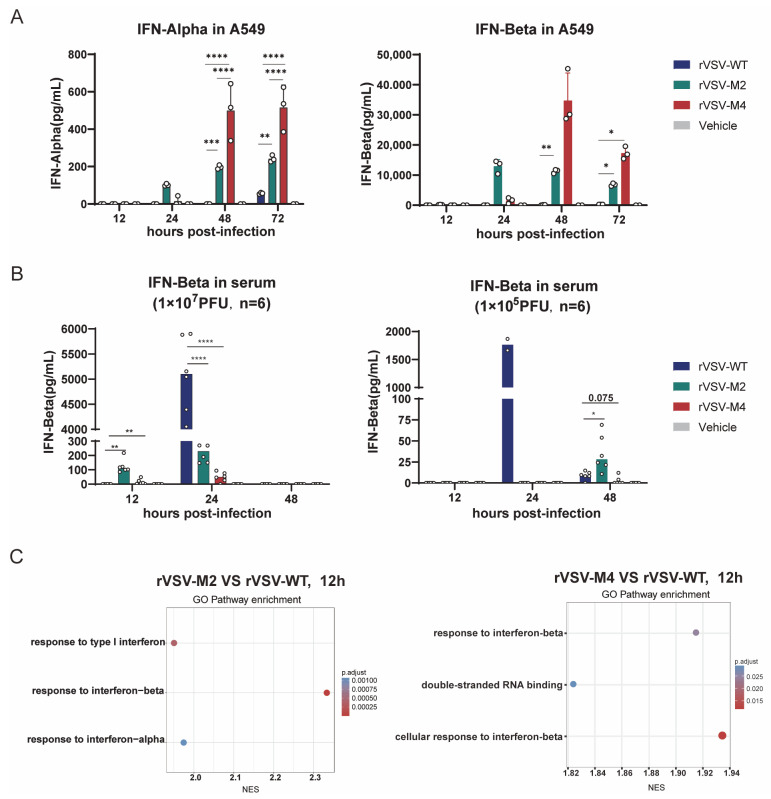
Type I IFN expression enhancement by M protein mutant. (**A**) Huh-7 and A549 cells were infected with rVSV-WT, rVSV-M2, or rVSV-M4 at MOI of 0.0001. The supernatants were harvested at 12, 24, 48, and 72 h post-infection to detect the levels of INF-α and INF-β. In A549 cells, the Mmut group exhibited the strongest interferon response, whereas the WT group showed the weakest. In Huh-7 cells, INF-α and INF-β levels in all groups were below the detection limit (Appendix A). n = 3; *, *p* < 0.05; **, *p* < 0.01; ***, *p* < 0.001; ****, *p* < 0.0001. (**B**) C57BL/6 mice were immunized intranasally with two different doses of rVSV-WT, rVSV-M2, or rVSV-M4 at a single dose of 1 × 10^7^ PFU (left) or 1 × 10^5^ PFU (right). INF-β levels in serum were detected at 12, 24, 48, and 72 h post-infection. n = 6; *, *p* < 0.05; **, *p* < 0.01; ****, *p* < 0.0001. (**C**) C57BL/6 mice were immunized intranasally at a single dose of 1 × 10^5^ PFU. Mice were sacrificed at various time points., and whole lung tissues were collected for transcriptome sequencing. Differentially expressed genes were enriched and subjected to functional analysis.

**Figure 4 viruses-17-01062-f004:**
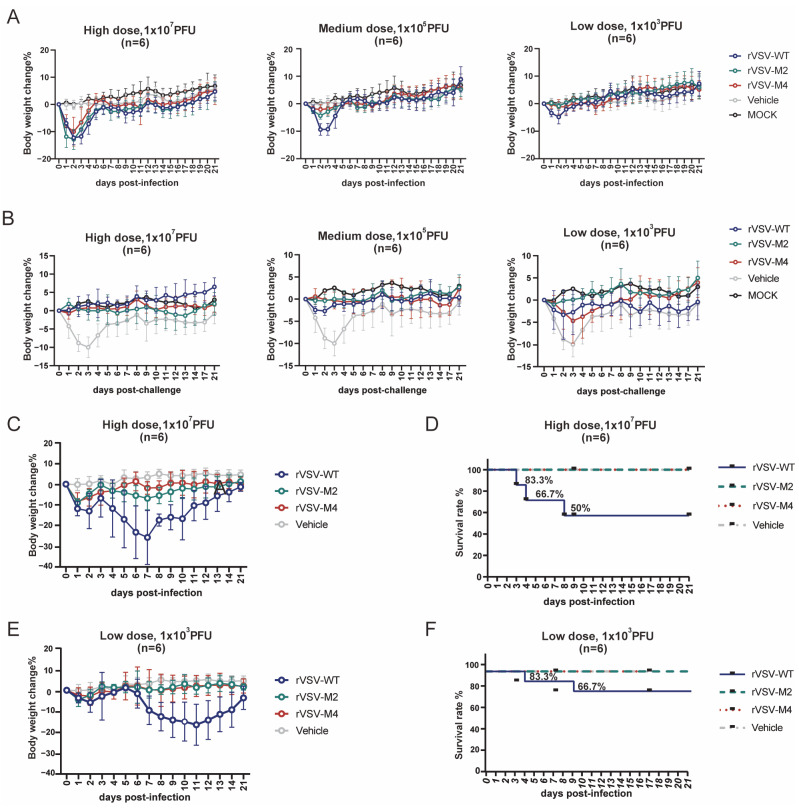
Comparison of the safety evaluation via intranasal and intracranial inoculation. (**A**–**F**): Female C57BL/6 mice aged 6–8 weeks were selected for immunization experiments. n = 6. Data are presented as mean ± standard deviation (SD). (**A**) Primary immunization: C57BL/6 mice were intranasally inoculated with a high (1 × 10^7^ PFU), medium (1 × 10^5^ PFU), or low (1 × 10^3^ PFU) dose of rVSV-WT, rVSV-M2, or rVSV-M4, respectively. The vehicle group received PBS, while the MOCK group remained untreated. Body weight changes were monitored over a 21-day period. Mice infected with rVSV-M2 or rVSV-M4 exhibited less weight loss compared to those infected with rVSV-WT. Less weight loss for M2 and M4 viruses only occurred at the 1 × 10^3^ and 1 × 10^5^ pfu experiments because at 1 × 10^7^, all VSV vaccinated mice lost weight early after vaccination. (**B**) Challenge post-immunization: Immunized C57BL/6 mice were challenged with a dose of 1 × 10^7^ PFU of rVSV-WT, while the MOCK group remained untreated. Body weight changes were recorded over 21 days. The protective efficacy of the M protein mutant rVSVs was comparable to that of rVSV-WT in C57BL/6 mice. (**C–F**): C57BL/6 mice were intracranially inoculated with high (1 × 10^7^ PFU) or low (1 × 10^3^ PFU) dose of rVSV-WT, rVSV-M2, or rVSV-M4, respectively, while the vehicle group received PBS. Body weight changes and survival rates were monitored over a 21-day period.

**Figure 5 viruses-17-01062-f005:**
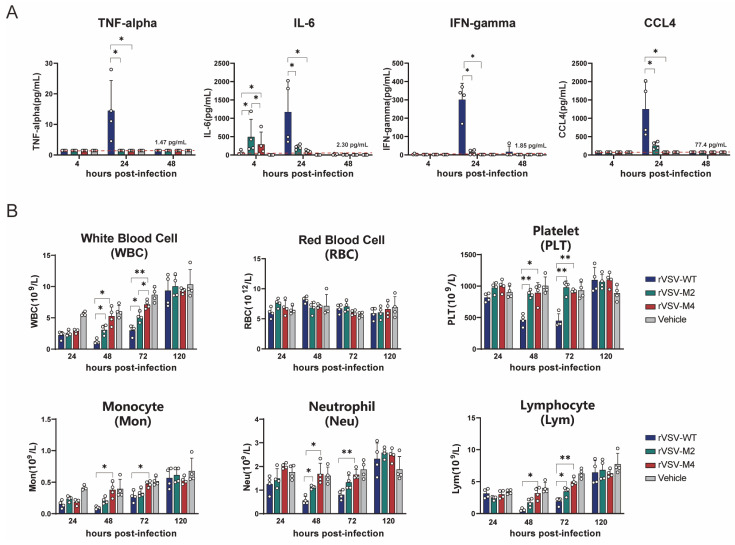
Comparison of inflammatory responses induced by rVSV-WT, rVSV-M2, or rVSV-M4 in vivo. *, *p* < 0.05; **, *p* < 0.01. (**A**,**B**): Female C57BL/6 mice aged 6–8 weeks were selected for immunization experiments. C57BL/6 mice were intranasally inoculated with a single dose of 1 × 10^7^ PFU of rVSV-WT, rVSV-M2, or rVSV-M4, respectively, while the vehicle group received PBS. Blood and serum were collected at various time points. n = 4. Data are presented as mean ± standard deviation (SD). (**A**) TNF-alpha, IL-6, IFN-gamma, and CCL4 levels in serum were detected at 4, 24, and 48 h post-infection. (**B**) Complete blood cell counts were performed to determine the numbers of white blood cells (WBC), red blood cells (RBC), and platelets (PLT). Additionally, white blood cells (WBC) were further classified into subpopulations.

**Figure 6 viruses-17-01062-f006:**
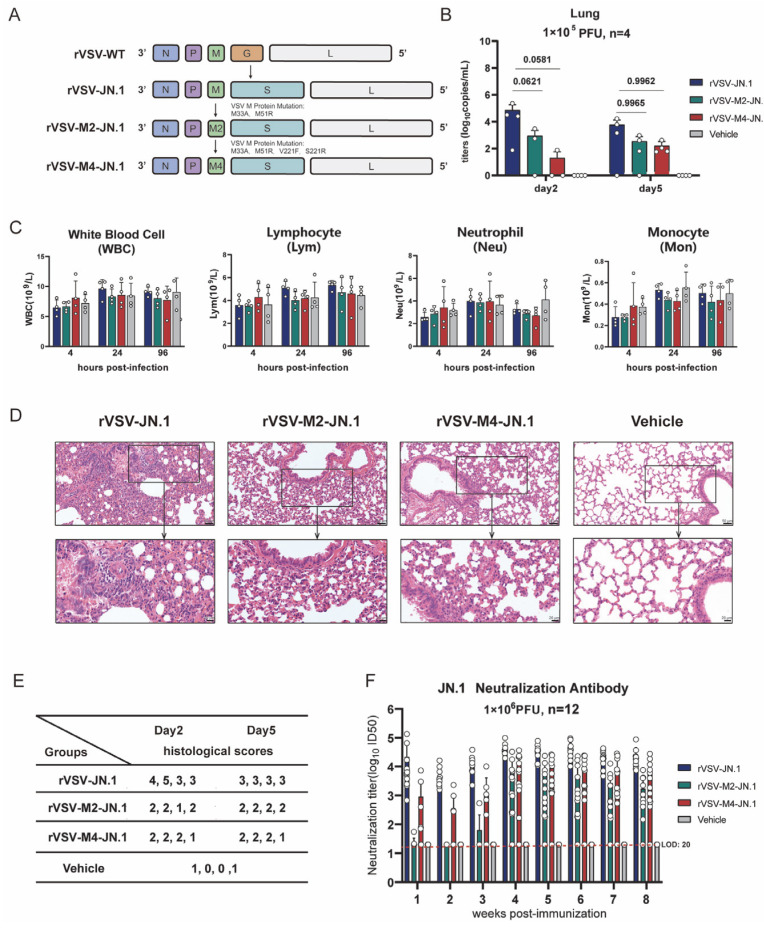
Construction of mucosal COVID-19 vaccine candidates based on rVSV. (**A**) Schematic diagram of viral construction: The VSV G protein was replaced with the COVID-19 spike protein through viral packaging technology, followed by point mutations in the VSV M protein. (**B**–**E**): Hamsters were intranasally immunized with a single dose of 1 × 10^5^ PFU, while the vehicle group received PBS. Blood and serum were collected at various time points, and hamsters were sacrificed on the second and fifth days post-immunization. n = 4. (**B**) The viral RNA titers in hamster lung tissue were quantified by RT-qPCR. (**C**) Complete blood cell counts were performed to determine and classify white blood cells. (**D**) Lung tissues in hamsters on day 2 and 5 post-infection were subjected to H&E staining to observe pathological changes. (**E**) Histological scoring of hamster lung tissue was conducted as follows: 0: Intact alveolar walls without thickening, no inflammatory infiltration, no congestion. 1: Mild diffuse inflammatory cell (neutrophil) infiltration in alveolar walls, no significant thickening. 2: Prominent and widespread inflammatory cell infiltration (neutrophils and monocytes), mild thickening of alveolar walls (1–2 times). 3: Severe inflammatory cell infiltration, localized thickening of alveolar walls (3–5 times). 4: Severe inflammatory cell infiltration, significant thickening of alveolar walls, 25–50% of lung tissue consolidated. 5: Severe inflammatory cell infiltration, significant thickening of alveolar walls, >50% of lung tissue consolidated. (**F**) Hamsters were intranasally immunized with two doses (1 × 10^6^ PFU per dose) of rVSV-JN.1, rVSV-M2-JN.1, or rVSV-M4-JN.1 on day0 and day21, while the vehicle group received PBS. Neutralizing antibody titers in serum were detected using a previously established pseudovirus neutralization platform. Nonlinear fitting was performed using GraphPad Prism 9.4.1 to calculate the ID_50_.

## Data Availability

The raw and processed data of RNA-seq experiment generated in this study have been deposited in the Gene Expression Omnibus (GEO) repository under SuperSeries accession code GSE296553. Any other raw data or non-commercial material used in this study are available from the corresponding author upon request. Source data are provided with this paper.

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
