# Peer review of "Development of COVID-19 Vaccine Candidates Using Attenuated Recombinant Vesicular Stomatitis Virus Vectors with M Protein Mutations"

_viruses, 2025, doi:10.3390/v17081062_

Round 1

Reviewer 1 Report

Comments and Suggestions for Authors

Recombinant vesicular stomatitis virus (rVSV) is a promising viral vaccine vector and continued research work is necessary considering the potential of new emerging pathogens so determining the benefits and shortfalls of the platform are critical. As such, the manuscript describes the potential of using previously known VSV M mutations for attenuating the virus and its use as a vaccine. For the most part, the manuscript is well written; however, the conclusions drawn by the experimental results are not fully supported by the data, which is a major issue. Additional research work is needed to strengthen the existing data and provide a clear and thorough evaluation of the proposed rVSV M mutant vectors. In some areas, the methodology and experimental designs should be explained in greater detail.

Line 2, Was the main goal of this study to develop COVID-19 vaccine candidates using attenuated (M mutants) VSV vectors? If yes, the title should reflect this.

Line 4, Please remove the word “and” between the 1 and *.

Lines 20-21, The following statement provides a clearer description, “We developed three candidate vaccines against SARS-CoV-2 using the wildtype VSV backbone with either wild-type M (rVSV-JN.1) and two M mutant variants (rVSV-M2-JN.1 and rVSV-M4-JN.1).”

Line 22, The enhancement of safety in hamsters was not fully evaluated. The H&E staining results aren’t sufficient to assess safety.

Line 52, Reference 22 doesn't appear to be appropriate to the text as there is no mention of a VSV-vectored EBOV vaccine in the reference as it is primarily discussing the structural differences in SARS-CoV-2 spike proteins.

Line 53, Please replace Ebola virus G protein with Ebola virus glycoprotein

Line 53, Please replace toxicity with neurotoxicity.

Line 59, Please replace Yong Kea with Yong Ke

Line 60, Please change attenuated rVSV to attenuated recombinant VSV (rVSV).   

Line 69, The extent of exploring viral pharmacokinetic distribution of the VSV vectors was very limited. Viremia and a full biodistribution study were not performed. The presence of the VSV-vectored viruses was shown only in the lung tissue and should be noted here.

Lines 69-72, As a suggestion for clarity lines 69-72 could be reworded to “More importantly, we developed three rVSV candidate vaccines against SARS-CoV-2 using the WT VSV M protein and two mutant M variants thus confirming viruses with M protein mutations retain strong immunogenicity while enhancing safety in hamsters.”

Lines 72-73, This appears to be the main goal of the study so the title should reflect this.

Lines 75-76, Since the COVID-19 pandemic is over it would be best to change “combat the COVID-19 pandemic” to combat any future COVID-19 outbreaks.”

Lines 85-88, Please reword “We constructed two VSV M gene fragments, designated as pVSV-FL-M2 and pVSV-FL-M4 respectively. The VSV M gene fragments and plasmid pVSV-FL-M2/M4 were digested with XbaI and MluI (NEB).”  It is a bit confusing and missing details. Based on the rescue strategy in the next section pVSV-FL-M2 should be a full length VSV plasmid encoding all VSV genes but here it is called a VSV M gene fragment. What is the difference between VSV M gene fragments and plasmid pVSV-FL-M2/M4?

Line 92, Please replace infiltrated with coated.

Line 93, Please remove atmosphere after 5% CO2

Line 95, Please provide the source for the recombinant vaccinia virus (vTF7-3).

Line 96, Please remove atmosphere after 5% CO2

Line 98, Regarding “pBS-VSV-N, 1 μg of pBS-VSV-P, 1.6 μg of pBS-VSV-G and 0.2 μg of pBS-VSV-L”, are these pBluescript backbone plasmids with a T7 promoter inserted because pBluescript plasmids are used in cloning and don't have a T7 promoter? If yes, a reference is needed on how they were made and discussed here.

Line 100, Please elaborate how the vTF7-3 virus was discarded from the infected cells.

Line 104, Please elaborate on why additional VSV-G was transfected into Vero cells. I presume the plasmid is under a T7 promoter like previously used so how is more VSV G expressed? Is the promoter under a CMV promoter? In Figure 1, the legend has the plasmid written as pCAG-VSV-G. Please clarify and provide more detail.

Line 106, How long did it take the Vero cells to show CPE? Was the supernatant harvested when more than 90% CPE? Please provide more details.

Lines 108-109, Regarding “optimally diluted”, what pfu/mL range would be considered optimally diluted? The expected plaque range is probably not known so were the samples serially diluted, and all dilutions plated to see the results? Please provide more details.

Line 110, Please change “4% formaldehyde solution was used to cells” to 4% formaldehyde solution was used to fix the cells

Line 110-111, How many days post-infection until the plaques were at the appropriate size?

Lines 113-114, How was the automatic intelligent imaging system optimized to accurately distinguish from plaques to cell monolayer aberrations? This should be mentioned in the description.

Line 116, Please explain why RT-qPCR was used for determining the virus titer (this was used in the lung tissue) and not the 6-well plaque assay? Many defective particles are generated during VSV infection so titers can be a log higher by using RT-qPCR versus the standard plaque assay thereby biasing the results.

Line 120, Please provide more details on the RT-qPCR method. What viral gene were the forward and reverse primers targeting?

Line 123, To generate the standard curve what viral gene was used as the standard?

Line 127, Please add the unit of measure for the virus titer - pfu/mL

Line 132, Please define the genetic makeup of these new psuedovirus. There are 3 variants with different VSV- M mutations.

Line 133, Please elaborate where these samples containing antibody came from?

Line 133, How much pseudovirus was used during the neutralization study?

Line 139, Please change CO2 with CO2

Line 137, How much lung tissue was sent for the RNA sequencing? Please provide more technical details about this method.

Line 141, Please provide more technical details about this method

Line 142, What specific kit from R&D systems was used for the interferon detection?

Line 143, Please define what was an optimal dilution. How was this determined?

Line 144, What absorbance was used for the detection?

Line 152, Please provide more detail to this section. Example: How were the animals vaccinated? How were the vaccines prepared?

Line 153, How many animals per group?

Line 159, Please provide more technical details for this section. What tissues were collected?

Line 180, Please change virus packaging technology to virus rescue technology as it is more appropriate to the generation of rVSV.

Lines 181-182, The passaging in BHK-21 cells and plaque purification step were not described in the Materials and Method section for the generation of the rVSV vectors. Please describe this. How many plaques were picked and expanded? How many passages were conducted? Please describe the rationale for passaging the cells in BHK-21 cells and plaque purifying the virus in Vero cells.

Line 182, Please change VERO to Vero

Line 183, Please change passage to infection

Line 184, Please change occurred to remained

Line 184, Please change viral to VSV

Line 184, How was the viral RNA extracted?

Line 185, Please add purification after agarose gel electrophoresis

Line 186, Please change gene to mutations

Line 187, What MOI was used to infect the BHK-21 cells?

Line 189, SDS-PAGE and Western blot method were not described in the Materials and Method section. Please include this information.

Line 191, Based on what was described, I am not sure if the SDS-PAGE can resolve any molecular weight differences between the WT and mutant M proteins with 2-4 point mutations. The level of expression would be more critical. Were the samples normalized to pfu values? If a sample has more viral particles (pfu/mL) than another sample, it might appear the expression level differs between the 2 viruses giving a false expression level.

Line 191, Please change content to level.

Line 193, Please provide more detail in this section. Titer values are not written out and only shown graphically.

Line 197-198, What MOI were the Huh-7 cells infected with? This should be mentioned here even though the results are similar to the BHK-21 cells.

Line 199, Please change detected through plaque assay to quantitated by the plaque assay.

Line 199, For Figure1E-G, it might be best to have a new figure (Figure 2) since viral replication rates are a separate topic and section.

Line 201, Please change “virus titers were not significantly different in BHK21 cells” to virus titers for Huh-7 cells (data not shown) were not significantly different from BHK21 cells

Lines 202-203, I would also add that the BHK-21 cells have an interferon defective pathway and had no titer difference between the 3 viruses supporting the sensitivity of the M mutants to interferon.

Line 205, In Figure 1F, the A549 image shows the plaque size difference between the 3 viruses but doesn’t show the greater log titer difference by the number of plaques in the image. Please explain. The supplementary S1 image shows the titer difference by CPE amongst the 3 viruses.

Line 205, The graph in Figure 1G shows the average plaque area but the text does not provide numerical values. Please include especially for the A549 cells since these cells showed the greatest difference.

Line 207, Please change VERO to Vero

Line 211, Please change “higher safety than rVSV-WT in vitro” to appear to be more attenuated than rVSV-WT in in vitro. This is not a safety indicator.

Line 213, Please change detect interferon responses to evaluate the interferon responses between the three rVSV viruses.

Line 218, What is the LOD in this assay since Huh-7 cells were below the LOD?

Line 219, Please provide numerical values to the interferon response between the 3 viruses.

Line 219, Please change “be enhanced” to “be the highest”

Line 220, What volume was used during inoculation? Please include the number of mice per group vaccinated.

Line 225, Why was IFN-beta only analyzed in the serum? IFN-alpha was measured in the in vitro experiments (Fig 2A), but IFN-alpha data is not presented here in the in vivo study. Please explain.

Line 227, Please provide numerical values of IFN-beta in the discussion. What is the LOD of the assay? This should be added to the figure.

Line 228, Looking at the graph, there appears to be only 2 points above zero, so IFN-beta appears to be very low for the M4 virus. This difference should be highlighted in the text as the response is not the same for the M2 virus.

Line 230, The rVSV-WT IFN-beta data doesn’t support the hypothesis that the M mutants had a reduced replication rate in mice because of the increased expression of IFN. First, the A549 data showed that IFN-alpha/beta were strongly stimulated for the M mutant viruses but not for the WT virus. Granted in vivo the situation is more complicated than an in vitro system. If the M mutant virus replication was hampered by a type 1 interferon response, then you would expect high levels of IFN detected in the M mutant serum samples; however, IFN is marginally detected in the serum samples. Please explain. If the M mutants can induce a faster stronger response, why isn't it detected?

Line 231, Please add virus before groups

Line 234, Please change transcriptomic to transcriptome. Please add more technical details of this method to the Materials and Method section.

Line 236, Why is a stronger type 1 interferon response not seen in Figure 2B for the M mutants? I do believe the M mutants would stimulate a stronger innate response, but the ELISA data does not show this. IFN-beta is barely detected in the M mutants. Please explain. The WT VSV vector shows a stronger IFN-beta response so you would expect a diminished replication, but this is not the case.

Lines 237-239 contradict the statement on Lines 230-231, rVSV-WT group exhibited a significantly stronger INF-β response than that of the attenuated groups. Please explain.

Line 246, Please add a statement on how the vaccines were formulated, i.e. diluted in PBS, dose volume (µL).

Lines 247-249, The stated conclusion for the weight loss study is too vague/general and doesn't accurately describe the results. Please add more detail. In the High dose group, all vaccinated mice except for the PBS mice showed rapid weight loss soon after vaccination with most of the weight loss lasting for 2-3 days post vaccination. With the error bars being quite large it is difficult to say the rVSV-WT vaccinated mice had the greatest drop. The weight rebounds around Day 6 post vaccination for the rVSV-M4 virus while the weights for the rVSV-WT and -M2 viruses seem to fluctuate more during the experiment. For the medium and low dose groups, the WT mice had the most drastic weight change out of all the groups for a duration of around 3-5 days post vaccination. However, by Day 6 the rVSV-WT mice weights were similar to the other VSV vaccinated mice. By Day 21, all VSV vaccinated mice have similar weights to the PBS control group. Has a statistical analysis been conducted to see if there were statistical differences between all vaccinated and dose levels considering the error bars are quite large?

Line 250, At Day 21, have you measured the antibody response against VSV G in all the groups prior to the challenge? If not, why weren’t antibody titers measured as this information would be beneficial.

Line 251, What was the route of challenge?

Line 255, Please define protection. Weight loss should not be the only factor in determining protection.

Lines 256-257, Are there any antibody data available otherwise you can’t say M mutants stimulated a strong protection? Weight loss data alone isn't strong enough evidence for the level of protection when the data points have a large error bar and weight fluctuations.

Lines 258-267, In the neurovirulence (NV) study were the animals monitored daily for signs of clinical illness following inoculation, with increased observations during the critical period: two to six days post-inoculation? Observations should include appearance, food and water intakes, behavior, and clinical signs of increased cardiac or respiratory rates. Please add more details regarding the neurovirulence study.

Line 259, A more stringent NV test is done with 5-6 week old mice as they are more sensitive to VSV via the intercranial (IC) route since wild-type VSV is 100% lethal. What is the age of the mice?

Line 261, For Figure 3D and F, please provide the live/dead numbers and % survival for each vaccinated group. Example: 6 out of 6 mice survived giving a survival rate of 100%.

Line 265, Typically, wild-type VSV is 100% lethal in mice via the IC route. Please explain why there is a 60% survival rate in mice inoculated with 1E+07 pfu as it should be 100% lethal? Based on the unusually high survival rate with WT VSV, the LD50 needs to be determined. Yes, the M mutations do appear to provide protection to mice via the IC route; however, the results are not indicative of a sensitive test due to the high survival rate, so it is difficult to say the M mutants have significantly reduced neurotoxicity. A statistical analysis needs to be conducted considering only an N=6 was conducted. A minimum of N=10 provides more power for such a critical experiment. The results show wild-type VSV is only 40% lethal at a high dose of 1E+7 pfu.

Line 269, What are the LODs for each measured cytokine?

Line 273-274, Please provide numerical values for the comparison.

Line 276, IL-6 in the M mutants peaked at 4 hours post infection (hpi) but by 24 hpi the values appear to drop by half. How is stability defined if the IL-6 values are diminishing as time progresses in the M mutants?

Line 285, Please change symptoms to blood cell counts.

Lines 287-288, By 48 hpi for the M4 group, WBC and monocyte levels were similar to the control but prior (24 hpi) the levels were lower. No changes in lymphocytes or neutrophils.

Line 288, It should be noted in the text that the PBS control group had also experienced elevated WBC, monocytes and lymphocytes at 120 hpi as compared to 24 hpi probably due to the IC inoculation. RBC, platelets, and neutrophils had no changes occurring.

Line 290, Yes, platelets were reduced at 48 and 72 hpi for the WT VSV group; however, at 24 and 120 hpi the platelet counts for the WT VSV group were not statistically different from the control thereby the thrombocytopenia is transient.

Line 292, A better descriptive would be less reactogenic than safety. Example: viral vaccines have been associated with thrombocytopenia, but this doesn't mean the vaccines are not safe. The MMR vaccine can cause this. I would stress that the M mutants were safer because WT-VSV can cause death via the IC route.

Line 293, same argument as in line 292. Reactogenicity is a better word.

Line 300, I presume the immunization of the hamsters intracranially is a typo and it should be intranasally.

Line 300, Why was a 1E+05 pfu dose chosen since previous studies used 1E+03 and 1E+07 pfu? The reason should be included in the text.

Line 302, Why weren’t other organs/tissues not tested for the presence of the rVSV vaccines? Yes, it is important to look at lung tissue especially after an IN vaccination but if the intent is to evaluate the attenuation nature of the virus it would be valuable to look at the potential dissemination of the virus into other tissues/organs. Measuring viremia is important as well. Since VSV G was replaced with the COVID-19 spike protein the tropism of the virus will change and should be evaluated.

Lines 303-305, This interpretation is not seen in the histology data in Figure 5D. The Supplementary S3 image is better.  All groups appear to be the same. Please point out the differences in the images.

Line 305, What is the WT group?

Line 305, Have you analyzed lung homogenates for live viruses via the plaque assay? If not, why wasn’t it?

Line 306-307, What is the LOD of the RT-qPCR assay. The Vehicle (PBS) control shows close to a 2-log titer, why is this? Need to subtract the background noise of the assay. Why only an N=4 for this experiment? More mice should have been tested for statistical significance. Have you looked at titers past day 5?

Line 309, Regarding “leading to enhanced safety compared to rVSV-JN.1.” Not enough research data is available to speculate that the virus vector is safer because it doesn't replicate well in vivo. Replicating less in vivo doesn’t equate to safety. With very limited immunogenicity data against the COVID-19 spike protein it is difficult to assess how well the M mutants have stimulated the immune system. The neutralization assay utilizes a pseudovirus system to measure neutralizing antibodies; however, neutralization assays in general may under or overestimate neutralizing antibodies. There are no significant indicators that the rVSV-JN.1 is less safe than the other viruses. rVSV-JN.1 may be more reactogenic than the other viruses.

Lines 313-316, Please clarify the attenuation modification for rVSV-JN.1 as stated.

Line 314, Please replace toxicity with reactogenicity.

Line 318, Do you mean the rVSV-JN.1 group when referring to WT group? Please change to rVSV-JN.1.

Line 319, Have you stained the lung tissues for granulocytes by immunohistochemistry?

Line 322, clinical scoring (Figure 5E). Histological changes are one thing but has this manifested in any clinical observations in the hamsters, i.e. ruffled fur, lethargy, elevated body temperature, changes in eating? Figure 5E is a histological or pathology scoring assessment and not a clinical assessment so please change.

Lines 326, Please replace safety with reactogenicity.

Lines 328, How many hamsters per group were vaccinated? Why were two doses at 1E+06 pfu chosen for this study?

Line 330, Why wasn’t anti-COVID-19 spike IgG antibodies measured? Or measuring secretory IgA antibodies in the mucosa of the upper respiratory tract.

Lines 322-333, Please expand on the results; provide numerical values. For example: For the rVSV-M4.JN1 group only after 3 weeks post-immunization did neutralizing antibodies appear. What is the LOD of this assay?

Lines 334-335, The presented data does not support this statement. rVSV-JN.1 may be more reactogenic, but it doesn’t appear less safe than the other viruses.

Line 336, Why are the graphs/ images in a separate section and not imbedded in the appropriate areas?

Line 343, Why wouldn't VSV be rescued at this point when the plasmids pVSV-FL-WT or pVSV-FL-M33A/M51R, along with helper plasmids (pBS-VSV-N, pBS-VSV-P, pBS-VSV-G, and pBS-VSV-L) were added? It’s not clear why the VSV G plasmid was transfected a second time.  pCAG-VSV-G is labeled differently in the Materials and Method.

Line 346, Methods were not discussed in Materials and Method. Please describe.

Line 350, the GraphPad Prism program is misspelled.

Line 350, Please move “GraphPad Prism was used to analyze the significant differences using the one-way ANOVA method. ****, p < 0.0001; ns, p>0.05.” as this is part of Fig 1E and not 1F.

Line 357, In Figure 2A and B, the "h's" after the numbers in the x-axis can be removed since the legend has hours post infection.

Line 361, Fig S1 is the CPE data and Fig S2 is the Huh-7 data and not BHK-21. Please correct.

Line 370, In Figures 3A, B, C and E the y-axis needs to be changed. Mice start at a 100 body weight % change which doesn't make sense. It should be a 0 with + values above the zero point and negative values below.

Line 376, It should be mentioned that less weight loss for M2 and M4 viruses only occurred at the 1E+03 and 1E+05 pfu experiments because at E+07, all VSV vaccinated mice lost weight early after vaccination.

Line 380, The IC data should be put in a separate figure as it is a different experiment than the challenge study.

Line 380, Shouldn’t it be intracranially inoculated instead of intranasally?

Line 380, Please include some statistical analysis comparing PBS (vehicle) controls with the vaccinated groups to see if the losses are significant. Example: 1 animal loss brings survival down to 83%; however, with the low N number the loss is not significant. Please provide the ratio of alive to dead such as 3 out of 6 = 50%, 4 out of 6 = 67%, 5 out of 6 = 83% and 6 out of 6 = 100%. This should be included in the text. The line data doesn’t appear to match.

Lines 380-382, For Fig 5C-F, the description lacks the explanation that this is a neurovirulence study.

Line 389, In Figure 4A and B, the "h's" after the numbers in the x-axis can be removed since the legend has hours post infection.

Line 390, After each name please add the abbreviation, i.e., white blood cells (WBC)

Line 399, In Figure 5B, an x-axis legend is needed here. The LOD should be mentioned and highlighted in the graph.

Line 400, In Figure 5C, an x-axis legend is needed here: hours post infection. The h’s can be removed.

Line 401, When the hamster lung tissues were harvested should be added to the description.

Line 409, In the Figure 5F top heading, please define JN.1 PsV in the text. Please add the LOD value.

Line 422-424, The COVID virus is a respiratory virus so administrating a vaccine via IN may provide a benefit due to the local stimulation of the immune system at the site of infection; however, the selected reference does not compare IN vs IM vaccination so difficult to make a statement that a "VSV-based COVID-19 vaccine is more suitable for nasal administration rather than intramuscular route."

Line 439, For clarity, please change in vitro to as seen by the in vitro data.

Line 449, It should also be noted in the text that the overall IFN-beta response was significanlty higher for the rVSV-WT virus at 24 hpi than the M mutants.

Line 451-453, It should also be noted in the text that the M protein mutant viruses have a reduced replication rate as this is supported by the in vitro A549 data.

Line 461-463, The selected reference is regarding a VSV-based COVID-19 vaccine in a phase I clinical trial administered via IM which did not show a robust immune response in people. One cannot make a general statement from this study that IN vaccination is better than IM because this can be specific to the COVID-19 spike antigen as there are countless VSV-based vaccines administered IM that show robust immune responses. Also, many vaccine designs have shown promising results in animals but fail in people. The provided reference only supports that a COVID-19 vaccine administered via IM is not a suitable route and IN may be a potential route but until phase I data is generated it is all speculative.

Line 469, Figure 3F does not show 50% mortality in mice. It shows a 20% mortality. Please explain. How about the mortality data at the 1E+07 pfu dose?

Line 476-478, Please indicate where this data is. Neutralization antibodies against VSV G have not been measured, only mouse weight measurements. The statement is speculative.

Lines 484-486, This is only true for the M4 mutant VSV. Comparing the lymphocyte data between the rVSV-M2 and vehicle control (PBS) at 48 and 72 hpi, the lymphocyte values for rVSV-M2 were lower than the PBS control. It only rebounded at 120 hpi.

Lines 491-493, Based on the data in Figure 4A, TNF-alpha, IFN-gamma and CCL-4 for the rVSV-M2 and -M4 mutants did not show a significant response above background as measured. Please justify the statement “a quicker and stable antiviral response antiviral response”. Only rVSV-WT showed a significant signal above background.

Lines 496-498, This data alone can not constitute safety of the vaccine.

Line 500, Please change JN.1 to VSV-JN.1-Spike pseudovirus

Lines 500-501, The generation of neutralization titers doesn't constitute improved safety.

Lines 513-515, The various point mutations in VSV M have been previously identified and their effects known, please describe what is novel in this study, i.e. combining the 4 point mutations in one vector.

Lines 516-521, Confusing statement as it mentions that a replication-defective vector is safer and a more promising vaccine vector while the data in the manuscript is with a replication competent vector. The future plan to conduct more in-depth research on a replication-defective rVSV takes away on the data generated in this manuscript. Does this mean that the M2 or M4 M mutants would be abandoned?

Lines 526-528, Only the neurovirulence mouse study was assessing the safety of the vectors. The other experiments were comparing the reactogenicity of the M mutants with the wild-type VSV.

Lines 527-528, Please expand. Too vague regarding the evaluation of the attenuation mechanism.

Line 530, No COVID-19 challenge data, so efficacy was not shown.

Lines 532-535, Confusing statement, please remove.

Reviewer 2 Report

Comments and Suggestions for Authors

In this paper, the authors have developed two recombinant VSV bearing two or four mutations in the M protein, respectively. Additionally, they have created three VSV-vector-based vaccines that express the spike protein of SARS-CoV-2. The in vitro and in vivo assays indicate that these recombinant viruses may be promising candidates for a mucosal vaccine against SARS-CoV-2. The results are convincing, and the paper is well written. I have only minor concerns.

Introduction

  • Page 2, line 59: The reference 27 belongs to Yong Ke.

Materials and Methods

  • The construction of the three vaccine candidates is not detailed in the Method section. Additionally, the description of the selected SARS-CoV-2 variant, JN.1, is necessary.
  • The pseudovirus-based neutralization assay (2.7) is not clear; please, provide a brief explanation.
  • The method for detecting type I interferon (2.9) should be briefly described.
  • The ELISA test for INFb quantification should be briefly explained in the Methods section.
  • Detailed information regarding the western blot assay and the antibodies utilized must be included in the Materials and Methods section.

Results

  • Fig 2C, the data presented for rVSV-M2 (“response to type I interferon”, “response to interferon beta”, “response to interferon-alpha”) differ from that of rVSV-M4 (“double-stranded RNA binding”, “cellular response to interferon beta”). Please, clarify or unify.
  • Legend to Figure 3 (page 10, line 380). The term "intranasally" should be replaced with "intracranial," as specified in the text on page 6, line 260.
  • Similarly, in the legend to Figure 4 (page 11, line 386), "intranasally" should also be replaced with "intracranial" in accordance with the text on page 6, line 280.
  • Figure 6A, the legend at the right of the image is left.

Discussion

  • Page 13, lines 458-460. The authors mention that the M protein mutant rVSV can replicate rapidly in tumor cells, indicating its potential as an oncolytic virus. Please provide relevant references to support this statement.
  • Page 14, lines 511-513. Please, provide references that support the assertion that the M51R mutation enhances type I interferon production during VSV infection.
  • Page 15, lines 532-535 should be omitted.

References

  • The reference to Menicucci et al., (Reference 12) is not in the correct format.

Reviewer 3 Report

Comments and Suggestions for Authors

Authors attempt to manipulate VSV M protein, an important virus for swine and ruminant vesicular disease, for a vector to apply on the development of COVID-19 vaccine in hamster model.

General comments: Authors are sound in lab technology training, but trying to  present too much at one time.  Some details disrupt the flow of manuscript reading.

specific comments:

line 2: the title is incomplete.  it does not cover the later half of the manuscript.

line 4: the author listing is incomplete.

section 2.2 and 2.3: figure 1 should be cited here.  I suggest figure 1 split into two figures.  Figure 1A to 1E brought forward.

line 130: replace "preliminary" with "previous".

Section 3.1 : Figure 1A to 1E should immediately follow the text.

Section 3.2: Figure 1F and 1G (preferred separate from above) should immediately follow the text.

Section 3.3: Figure 2 should immediately follow the text.

line 213-217: this should be placed in M and M section.

Section 3.4 and Figures 3 and 4: To keep the safety issue in mind is good. The intranasal immunization is good.  But the major biohazard concern of VSV is on swine, ruminants, and humans to which VSV can infect naturally.   I would suggest delete this part or put this part in the supplementary, and keep only those related to intranasal immunization in the text.  This section occupies too much space and interrupt the reading to Figure 5.

section 3.5: Figure 5 should immediately follow the text.

Discussion: discuss more on the pathogenesis.

Round 2

Reviewer 1 Report

Comments and Suggestions for Authors

Thank you for addressing the comments and making the appropriate changes.

Reviewer 3 Report

Comments and Suggestions for Authors

Most of my comments have been addressed. The discussion has improved.